# Long Non-Coding RNA as a Potential Biomarker for Canine Tumors

**DOI:** 10.3390/vetsci10110637

**Published:** 2023-10-30

**Authors:** Yan Zhang, Meijin Wu, Jiahao Zhou, Hongxiu Diao

**Affiliations:** Key Laboratory of Animal Pathogen Infection and Immunology of Fujian Province, College of Animal Sciences, Fujian Agriculture and Forestry University, Fuzhou 350002, China; mszyzhangyan@foxmail.com (Y.Z.); 13338528861@163.com (M.W.); 18609533388@163.com (J.Z.)

**Keywords:** long non-coding RNA, biomarker, canine tumors

## Abstract

**Simple Summary:**

Cancer is the leading cause of death in dogs. Currently, researchers find long non-coding RNA (lncRNA) is a crucial regulator in the progression of various types of cancers. This article emphasizes the current and future advancements in lncRNAs as potential biomarkers for diagnosing, monitoring, and treating different types of canine tumors, which will benefit veterinary medicine as well as comparative medicine.

**Abstract:**

Cancer is the leading cause of death in both humans and companion animals. Long non-coding RNA (lncRNA) plays a crucial role in the progression of various types of cancers in humans, involving tumor proliferation, metastasis, angiogenesis, and signaling pathways, and acts as a potential biomarker for diagnosis and targeted treatment. However, research on lncRNAs related to canine tumors is in an early stage. Dogs have long been considered a promising natural model for human disease. This article summarizes the molecular function of lncRNAs as novel biomarkers in various types of canine tumors, providing new insights into canine tumor diagnosis and treatment. Further research on the function and mechanism of lncRNAs is needed, which will benefit both human and veterinary medicine.

## 1. Introduction

Cancer is the leading cause of death in both humans and companion animals. Approximately 19.3 million new cancer cases and almost 10.0 million cancer deaths occurred in 2020 [1]. According to Small Animal Clinical Oncology, 23% of companion animals undergoing necropsy died of cancer without age adjustment [2]. As the main companion animal, dogs are aging, and the occurrence of tumors is also increasing; it has become one of the leading causes threatening the lives of middle-aged and elderly dogs. Research has found that 50% of dogs aged 10 or more years old succumb to cancer, and one in four dogs will develop cancer at some point in their lives [3]. Additionally, cancer remains the most concerning health issue for 41% of dog owners [4]. Despite advances in treatment with a combination of surgery and/or chemotherapy increased over the past 15 years, the prognosis remains poor due to the late diagnosis and metastasis.

Dogs are excellent models for comparative oncology due to the similarities between dogs and humans in terms of their living environment, spontaneous development, subtype classification, genomes, mutational profiles, and chromatin map overlaps [5,6]. Moreover, canines and humans share the same genes and pathways involved in tumorigenesis; for example, matrix metalloproteinase-9 remarkably increases cell malignancy via TGF-β/SMAD signaling [7]. In addition, dogs can be utilized to evaluate novel biomarkers or anticancer targets before clinical translation, which leads to significant growth in comparative oncology research encompassing from basic science to clinical trials [5]. Comparative oncology research as a new perspective on cancer has provided new sights in cancer research in the past decades. The use of biomarkers—for example, ER, PR, and Her 2 in mammary tumors—represented great value in human cancer diagnosis and treatment protocols [8]. However, no ideal biomarkers have been identified in veterinary medicine due to a lack of specificity and sensitivity. Hence, the research on biomarkers of canine cancer will not only benefit the veterinary clinic but will also provide advantages to humans.

Long non-coding RNA (lncRNA) is a class of heterogeneous RNA that lacks protein-coding capacity and the ability to produce functional small peptides [9,10]. Initially, lncRNAs were considered transcriptional noises [11], while the past decade has witnessed a recognition of their remarkable biological functionality, such as their ability to regulate transcription, epigenetic modifications, chromatin structures, and translation through the interactions with DNA, RNA, proteins, and even signaling receptors [12,13,14,15]. lncRNA can function as guide molecules, decoys, sponges for associated miRNA, or scaffolds to regulate various biogenesis processes (see Figure 1) [16]. In addition, lncRNA generally has lower expression than protein-coding genes and exhibits a high degree of tissue specificity [17]. Compared to human cancer, the research on canine lncRNAs began later. In 2014, Hoeppner et al. first established approximately 7200 lincRNA transcripts and 4600 antisense lncRNAs [18]. Then, Wucher et al. used FEELnc, a program that accurately annotates lncRNAs, to annotate 10,374 novel lncRNAs and 58,640 mRNA transcripts from 20 canine RNA-seq samples produced by the European LUPA consortium [19]. Another study has characterized the expression profiles of 10,444 canine lncRNAs in 26 distinct tissue types and found that 44% of canine lncRNAs are exhibited in a tissue-specific manner [20]. Further analysis revealed that the SINEC_Cf family can change the promoter sequence, which contributes to the spatial expression of canine lncRNA [20]. These studies demonstrate that lncRNA holds significant potential as a tumor diagnostic and prognostic marker. Here, we aim to highlight the molecular function of lncRNA as novel biomarkers in various types of canine cancer, as well as their relevance to current clinical practice.

## 2. The Molecular Function and Clinical Use of lncRNA as Canine Mammary Tumor Biomarkers

Mammary gland tumors are the most frequently diagnosed tumors in dogs (9–11 years old), with approximately 50% being malignant, which represents a significant clinical concern [21]. Surgery remains the primary treatment for all dogs diagnosed with mammary gland tumors, excluding inflammatory carcinomas and distant metastasis carcinomas. However, 58% of dogs developed a new tumor in the ipsilateral mammary chain after the initial surgery, and 77% required additional surgery [22]. Chemotherapy is one of the major options for human mammary tumors, like cyclophosphamide, 5-fluorouracil, tamoxifen, paclitaxel, etc. [23]. In veterinary, chemotherapy does not offer long-term improvement and leads to obvious side effects and chemotherapy resistance. Early diagnosis via a sensitive biomarker of the canine mammary tumor can alter this unfortunate situation.

In recent decades, the use of biomarkers (like CA15-3, COX-2, BRCA1, and BRCA2) in canine mammary tumors has been limited due to a lack of sensitivity and specificity [24,25]. lncRNAs may exhibit strong promise as novel biomarkers due to high tissue specificity [20]. Lu et al. analyzed the expression profiles of lncRNAs in canine mammary tissues and adjacent non-tumor mammary tissues and identified 68 significantly differentially expressed lncRNAs. Further investigation revealed that a novel lncRNA named lncRNA 40589 inhibits cell proliferation, migration, and invasion. In contrast, the effects of lncRNA 34977 are opposite in both in vitro and in vivo [25]. These results indicate that lncRNA can function as a promoter or an inhibitor of tumor progressions. Another study has shown that lncRNA 34977 promotes the development of canine mammary tumor cells and suppresses apoptosis by directly targeting miR-8881/ELAVL4 [26]. Chemotherapy resistance is a significant factor contributing to the failure of canine mammary tumor treatment. Xu et al. screened lncRNAs associated with tamoxifen resistance in canine mammary gland tumor cells and found a novel lncRNA 42060 significantly upregulated in drug-resistant cells and tumor tissues [27]. Further investigation revealed that lncRNA 42060 functions as a sponge for miR-204-5p regulating SOX4 expression activity, leading to the inhibition of tumor cell proliferation, migration, clone formation, restoration of tamoxifen sensitivity, and a reduction in stem cell formation in drug-resistant cells [27]. These studies demonstrate that lncRNAs hold great promise as innovative biomarkers and therapeutic targets for canine mammary tumors.

## 3. The Molecular Function and Clinical Use of lncRNA as Canine Melanoma Biomarkers

Melanoma occurs frequently in dogs, accounting for 5–7% of all canine neoplastic diseases [28]. Canine melanomas are characterized by high invasiveness and a propensity for metastasis, with a one-year survival rate of about 20% when treated with surgery alone [29,30]. Oral melanomas exhibit the worst prognosis with a 10% one-year survival rate [31]. Mucosal melanoma is the predominant form of malignant oral tumors with extremely aggressive behavior and tends to occur in breeds such as the Cocker spaniel, miniature poodle, Anatolian sheepdog, Gordon setter, Chow Chow, and golden retriever [32]. Interestingly, specific breeds with mucosal melanoma have similar clinical features and histopathology striking homologies with human melanomas, which allows canine melanomas to serve as spontaneous models for human melanoma, particularly for non-UV-induced melanomas [30,33]. Early diagnosis contributes to the long-term survival of oral melanoma patients. However, most dogs with oral malignant melanoma are diagnosed at an advanced stage due to non-obvious clinical signs, like increased salivation, epistaxis, halitosis, and dysphagia, which are often mistaken for dental disease, or even neglected. Additionally, the absence of biomarkers limited the diagnosis and monitoring of melanoma progression.

In recent years, deep transcriptome sequencing has been conducted in canine melanoma. In 2019, 417 differentially expressed lncRNAs were annotated in canine mucosal melanomas in comparison with matched control tissues [34]. Further comparative genomic analysis with this set of data revealed that 26 of these lncRNAs were reported to be conserved in humans, such as lncRNA SOX21-AS, lncRNA ZEB2-AS, and lncRNA CASC15 [34]. Unsupervised co-expression network analysis with coding genes suggested that these lncRNAs function as regulatory elements in the cell cycle and carbohydrate metabolism. Finally, this study points to the potential for using lncRNAs as diagnostic or therapeutic targets for canine oral melanomas [33]. Unfortunately, the details of the mechanism are still unclear. In another study, the analysis focused on novel small non-coding RNAs found in exosomes of two melanoma cell lines (KMeC and LMeC). The study identified 55 exosomal lncRNAs that were differentially expressed and lncRNAs ENSCAFT00000069599.1 and ENSCAFT00000090032.1 verified significantly high expression levels in canine oral melanoma patients [35]. A particularly interesting finding is that lncRNA ENSCAFT00000069599.1 demonstrated the highest diagnostic efficacy, as determined by receiver operating characteristic (ROC) curve analysis of sensitivity and specificity [35]. This suggests that lncRNA ENSCAFT00000069599.1 has the potential to be a valuable diagnostic biomarker for canine oral melanoma. Combining these research findings sheds light on the diagnostic utility of lncRNAs as promising biomarkers for canine melanoma. However, the function and mechanisms of these founding lncRNAs are still unknown, and are in need of further exploration.

## 4. The Molecular Function and Clinical use of lncRNA as Canine Lymphoma Biomarkers

Lymphoma is a common disease in dogs—particularly prevalent in Labrador Retrievers, Rottweilers, and Boxers [36,37,38]—which displays an important comparative model for drug development with human lymphomas [39,40]. Chemotherapy remains the primary treatment approach for canine lymphoma due to the condition’s systemic presentation. In the past few decades, numerous single-agent and multi-agent chemotherapy protocols have been extensively studied. Single-agent doxorubicin is a relatively simple chemotherapy protocol with intermediate effectiveness, but cumulative cardiac toxicity is a significant issue in dogs [41]. CHOP (cyclophosphamide, doxorubicin, vincristine, and prednisone)-based protocols have been widely utilized. Although most dogs will experience a significant short-term remission, 95% of patients will relapse and eventually die [40]. The early diagnosis, prediction of the outcome, and treatment target are still challenging in canine lymphoma.

Omics have been utilized in profiling canine lymphoma and identifying potential biomarkers. In 2019, Aresu and colleagues first conducted an integrated analysis using transcriptome sequencing, genome-wide methylation analysis, copy number variation analysis, and clinical outcome data to comprehensively identify the molecular mechanisms of canine diffuse large B-cell lymphoma at the coding gene level. The study identified many individual transcripts and deregulated pathways, such as MYC signaling, PI3K/AKT/mTOR, and NF-κB affected by mutated genes [42], and these signaling pathways have been well studied in humans. A cross-species analysis of lncRNAs revealed a non-negligible fraction of human diffuse large B-cell lymphoma lncRNAs is also expressed in canine lymphoma. In particular, 244 human novel lncRNAs that lifted over to CanFam3 exhibited partial exonic overlap with dog transcripts [43]. Additionally, these lncRNAs were found to be predominantly situated in the same genomic regions [43]. These findings suggest that lncRNAs are related to the development of lymphoma both in dogs and humans and may be induced by the same mechanism in onco-pathology. Another study by Cascione developed a customized R pipeline to identify the expression of novel and annotated lncRNAs in diffuse large B-cell lymphoma to differentiate B-cell lymphoma subtypes. This methodology can classify diffuse large B-cell lymphoma into two main groups. Interestingly, these two diffuse large B-cell lymphoma groups exhibited statistical differences in mortality rates, indicating the potential of utilizing lncRNAs as predictive biomarkers during the time of diagnosis [44]. Taken together, these works support the translational application of lncRNA as a potential diagnostic biomarker for canine lymphoma, and even human lymphoma. Unfortunately, the function and experimental characterization of specific lncRNA in canine lymphoma have not been investigated.

## 5. The Molecular Function and Clinical Use of lncRNA in Other Canine Tumors

lncRNA is also associated with other types of cancer in canines. Telomeric repeat-containing RNA (TERRA) is a long non-coding transcript derived from telomeres, which regulates the growth of canine soft tissue sarcoma cells by inhibiting telomerase activity [45]. One well-studied long non-coding RNA, MALAT1, plays a crucial role in the regulation of multiple signaling pathways in osteosarcoma cells, involving tumor proliferation and metastasis, resulting in a low proliferation capacity. Notably, MALAT1 is highly expressed in two canine osteosarcoma cell lines, COS3600 and COS4074 [46]. Several studies have confirmed that MALAT1 plays a crucial role in the regulation of multiple signaling pathways in osteosarcoma cells, involving tumor proliferation and metastasis via sponging miRNAs [47,48,49]. These findings further demonstrate the presence of similar features between human and canine osteosarcoma, indicating that dogs serve as an ideal model for human cancer. The presence of tumorigenic lncRNAs was also explored in Madin-Darby canine kidney (MDCK) cells. The screening was conducted between low tumorigenicity and highly tumorigenic MDCK cell lines, resulting in the identification of 1092 annotated and 619 novel lncRNAs. Further investigation demonstrated that the lncRNA MSTRG.1056.2 can directly regulate ERBB3, triggering the activation of the PI3K-Akt pathway, which ultimately contributes to tumorigenesis [50]. In addition, lncRNA HULC, which is highly upregulated in liver cancer, promotes the differentiation of canine adipose-derived stem cells (ADSCs) into epithelial and smooth muscle-like cells via the upregulation of BMP9 activating Wnt/β-catenin signaling, while simultaneously deactivating the Notch signaling [51]. In canine histiocytic sarcoma, exhaustive genome-wide association studies (GWAS) identified a set of predisposing variants with regulatory effects in non-coding regions, which ultimately lead to strong cancer predisposition in specific dog breeds [52].

At present, several lncRNAs have been investigated in canine tumors for their potential as biomarkers in tumor progression. Table 1 summarizes the lncRNAs examined in veterinary experiments, including their names, functions, and mechanisms. However, the functions of most lncRNAs have not been characterized. Further investigations of these lncRNAs may provide amazing insights for oncology or other pathological areas in veterinary medicine.

## 6. Summary and Perspective

lncRNAs act as crucial regulators in the development of various types of cancers. This article summarizes the existing knowledge of lncRNAs in canine tumors (See Table 1) and sheds light on the functions of these lncRNAs as promising diagnostic or target treatment biomarkers. Transcriptome sequencing has identified a considerable number of abnormal lncRNAs in canine tumors, but there are a few lncRNAs that have been associated with canine tumors, and the mechanism remains unclear, which may delay the progress of the application of lncRNAs in veterinary clinics. Further studies to fully characterize the functions of annotated lncRNA, novel lncRNAs, as well as small peptides encoded by lncRNAs, will be highly beneficial. Additionally, detecting specific differential changes in lncRNAs and analyzing their correlation with tumor grading, staging, and prognosis in clinical cases will provide valuable insights for the clinical treatment of canine tumors. Over the past few decades, spontaneous canine tumors have been extensively used as models for human cancer clinical trials, so further in-depth research on the function and mechanism of lncRNAs, particularly these conserved lncRNAs with humans, will benefit both human and veterinary medicine.

## Figures and Tables

**Figure 1 vetsci-10-00637-f001:**
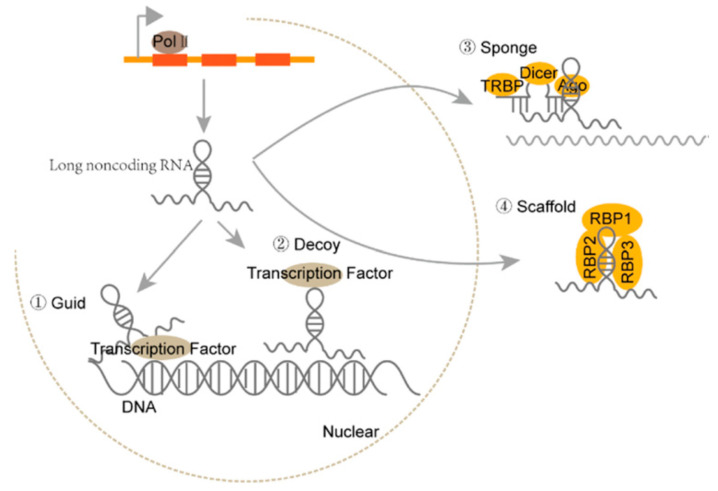
The biogenesis and effector types of machinery of lncRNAs [16].

**Table 1 vetsci-10-00637-t001:** List of lncRNAs and their role in canine tumor development.

Cancer Types	lncRNA	Function	Reference
Mammary tumor	lncRNA 40589	inhibits cell proliferation, migration, and invasion	[24]
	lncRNA 34977	promotes the development of tumor cells and suppresses apoptosis	[24]
	lncRNA 42060	inhibiting tumor cell progression	[26]
Melanoma	lncRNA SOX21-AS	regulatory elements in the cell cycle and carbohydrate metabolism	[33]
	lncRNA ZEB2-AS	regulatory elements in the cell cycle and carbohydrate metabolism	[33]
	lncRNA CASC15	regulatory elements in the cell cycle and carbohydrate metabolism	[33]
	lncRNAs ENSCAFT00000069599.1	correlation with the patient’s ROC	[34]
	lncRNAs ENSCAFT00000090032.1	NA	[34]
Lymphoma	A group of lncRNAs	a non-negligible fraction of human DLBCL	[42]
	A group of lncRNAs	differentiates B-cell lymphoma subtypes and indicates mortality rates	[43]
Soft tissue sarcoma	lncRNA TERRA	regulates the cell growth	[44]
Osteosarcoma	lncRNA MALAT1	regulates the cell growth	[45]
MDCK tumorigenicity	lncRNA MSTRG.1056.2	contributes to tumorigenesis	[49]

NA: Has not been tested.

## Data Availability

Not applicable.

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
