# Peer review of "Long Non-Coding RNA as a Potential Biomarker for Canine Tumors"

_vetsci, 2023, doi:10.3390/vetsci10110637_

Round 1

Reviewer 1 Report

The authors of the review publication " Long Non-coding RNA as A Potential Biomarker for Canine Tumors " conducted a detailed review of the world literature in English on LongNon-coding RNA (LN RNA) in both human and canine medicine. The purpose of the study was to determine the usefulness of LN RNA as a potential biomarker for canine cancer tumors and to shed light on the mechanisms of LN RNA involvement in tumorigenesis, Due to the considerable similarity of the mechanisms of human and canine tumor formation, such an analysis seems warranted to approximate these mechanisms in human medicine. However, the described review of the current information on this subject showed the enormous diversity of the LN RNA molecule going into the hundreds and the diverse mechanism of LN RNA involvement in both the formation and inhibition of cancerous tumors. The authors rightly point out in many places that further detailed studies of this molecule are needed to determine its usefulness as a cancer biomarker, but what I find most lacking in this publication is the authors' critical view of its usefulness as a biomarker. Hence, I ask you to complete and answer the following questions ;

1. Does the large variation in the structure of LN RNA allow it to be considered as a single molecule ?

2. Whether the participation of LN RNA in both the processes of stimulation and inhibition of tumor formation makes it a good candidate as a biomarker?

Please also add in the conclusion of the paper information, in the opinion of the authors, about the shortcomings and weaknesses of this publication and possible studies that should be carried out to consider LN RNA as a clinically reliable biomarker 

Author Response

We deeply appreciate your critical comments, which have undoubtedly provided us with valuable opportunities to improve our work. We have revised the manuscript following the advice provided and addressed the issues raised by the reviewer. The changes were highlighted in yellow in the manuscript. The following are point-by-point responses to comments and suggestions from authors:

1. Does the large variation in the structure of LN RNA allow it to be considered as a single molecule ? 

Response: We think that lncRNA can be used as a biomarker for tumors because lncRNAs have high tissue-specific in tumors. In addition, lncRNA can target miRNA, mRNA, and protein regulating tumor progress. Maybe like other biomarkers, several lncRNA will work better in tumor prognosis or treatment. There still needs more and more research.

2. Whether the participation of LN RNA in both the processes of stimulation and inhibition of tumor formation makes it a good candidate as a biomarker?

Response: In human cancer, some certain lncRNAs are carcinogenic, whereas others exhibit anti-cancer properties. Once we understand its function and mechanism, it can be as a biomarker for diagnosis, treatment, and prognosis assessment, like ER, PR, and Her 2 in human mammary tumors.

3. Please also add in the conclusion of the paper information, in the opinion of the authors, about the shortcomings and weaknesses of this publication and possible studies that should be carried out to consider LN RNA as a clinically reliable biomarker

Response: The main shortcomings of this article are that there are not a huge amount of references about lncRNAs in canine tumors. As you know the study of lncRNAs in canine cancer is much later than in human cancer, we need more time and effort to explore the crucial roles lncRNAs play in veterinary clinics. Accepting your comments, we added shortcomings of this article and possible studies that can be carried out in the future.

Reviewer 2 Report

Dear Authors, the review is fine but its originality could be improved. It would be useful to write a chapter that also addresses the topic "Long non-coding RNA as a potential relationship between biomarkers and epithelial-mesenchymal transition (EMT) in epithelial tumor metastases". Many articles on this topic have been published on humans, but currently none are related to the canine species. Enhancing this review with this topic would make it the first in the veterinary medical field to have addressed this very timely topic. 

Author Response

We deeply appreciate your critical comments, which have undoubtedly provided us with valuable opportunities to improve our work. We have revised the manuscript following the advice provided and addressed the issues raised by the reviewer. The changes were highlighted in yellow in the manuscript. The following are point-by-point responses to comments and suggestions from authors:

Dear Authors, the review is fine but its originality could be improved. It would be useful to write a chapter that also addresses the topic "Long non-coding RNA as a potential relationship between biomarkers and epithelial-mesenchymal transition (EMT) in epithelial tumor metastases". Many articles on this topic have been published on humans, but currently none are related to the canine species. Enhancing this review with this topic would make it the first in the veterinary medical field to have addressed this very timely topic.

Response: We deeply appreciate your critical comments. The role of EMT in tumor metastasis is undeniably significant, but unfortunately, there has been a lack of research on the involvement of lncRNA in EMT in canine tumors. Considering the focus of the journal Veterinary Science, it may be inappropriate to solely discuss lncRNAs in the context of EMT in human tumor metastases.

Reviewer 3 Report

Comments on vetsci-2620144-peer-review-v1 entitled “Long Non-coding RNA as A Potential Biomarker for Canine Tumors” by Zhang et al

The manuscript describes an interesting review on Long Non-coding RNA and its potential use as biomarker for tumor diagnosis and treatment. New articles in this theme are welcome. 

In general, the writing is clear. However, there are some changes that should be amended to increase comprehensibility.

Minor Changes

Page 3, lines 80-81: Please replace “… Mammary gland tumors (9-11 years old) are the most frequently diagnosed tumors in dogs, …” to “… Mammary gland tumors (are the most frequently diagnosed tumors in dog (9-11 years old), …”

Page 3, lines 95: Please replace “… adjacent non-tumor tissues, …” to “… adjacent non-tumor mammary tissues…” or “… adjacent non-neoplastic mammary tissues  …”

Page 3, lines 95: Please write “in vitro” and “in vivo” in italic.

Page 3, line 112: Please amend “…5%-7%...” to “…5-7%...”

Page 4, lines 175-176: Please clarify the sentence “…These findings suggest that the lncRNAs related to the development of lymphoma between dogs and humans may exist similarly even same mechanism.” Do authors mean “These findings suggest that lncRNAs are related to the development of lymphoma both in dogs and humans, and may be induced by the same mechanism??”

Page 5, lines 193: Please clarify the following sentence “…osteosarcoma cells, involving tumor proliferation and metastasis, is highly expressed in two canine osteosarcoma…”. Do you mean “osteosarcoma cells, is involved in tumor proliferation and metastasis, and is highly expressed in two canine osteosarcoma…??”.

Page 5, lines 193: Please clarify the sentence “…insights for veterinary clinics.” Do you mean “…insights for veterinary clinicians.”

Page 6, table 1: Please put a line between the head of the table (“Cancer types”, “lncRNA”, “Function” and “Reference”) and the body of the table (“Mammary tumor”, “Melanoma”, and so on…) to separate the fields.

Please write “Cancer types”, “lncRNA”, “Function” and “Reference” in bold.

Please replace “mammary tumor”, “melanoma”, and so on … to “Mammary tumor”, “Melanoma”, …

Dear Authors

In general the writing is clear, though some minor amends of English language are required.

Author Response

We deeply appreciate your critical comments, which have undoubtedly provided us with valuable opportunities to improve our work. We have revised the manuscript following the advice provided and addressed the issues raised by the reviewer. The changes were highlighted in yellow in the manuscript. The following are point-by-point responses to comments and suggestions from authors:

1. Page 3, lines 80-81: Please replace “… Mammary gland tumors (9-11 years old) are the most frequently diagnosed tumors in dogs, …” to “… Mammary gland tumors (are the most frequently diagnosed tumors in dog (9-11 years old), …”

Response: Accepting your comments, we replaced this sentence.

2. Page 3, lines 95: Please replace “… adjacent non-tumor tissues, …” to “… adjacent non-tumor mammary tissues…” or “… adjacent non-neoplastic mammary tissues …”

Response: Accepting your comments, we replaced “… adjacent non-tumor tissues, …” to “… adjacent non-tumor mammary tissues…”.

3. Page 3, lines 95: Please write “in vitro” and “in vivo” in italic.

Response: Accepting your comments, we corrected “in vitro” and “in vivo” in italic.

4. Page 3, line 112: Please amend “…5%-7%...” to “…5-7%...”

Response: Accepting your comments, we corrected “…5%-7%...” to “…5-7%...”

5. Page 4, lines 175-176: Please clarify the sentence “…These findings suggest that the lncRNAs related to the development of lymphoma between dogs and humans may exist similarly even same mechanism.” Do authors mean “These findings suggest that lncRNAs are related to the development of lymphoma both in dogs and humans, and may be induced by the same mechanism??”

Response: We were sorry to state this unclearly and corrected this sentence with your advice.

6. Page 5, lines 193: Please clarify the following sentence “…osteosarcoma cells, involving tumor proliferation and metastasis, is highly expressed in two canine osteosarcoma…”. Do you mean “osteosarcoma cells, is involved in tumor proliferation and metastasis, and is highly expressed in two canine osteosarcoma…??”.

Response: We were sorry to state this unclearly, and what we want to say is that lncRNA MALAT1 is highly expressed in two canine osteosarcoma cell lines (COS3600 and COS4074), and lncRNA MALAT1 plays a crucial role in tumor proliferation and metastasis. And we rephrased this sentence.

7. Page 5, lines 193: Please clarify the sentence “…insights for veterinary clinics.” Do you mean “…insights for veterinary clinicians.”

Response: We were sorry to state this unclearly. It should be veterinary medicine, and we corrected it.

8. Page 6, table 1: Please put a line between the head of the table (“Cancer types”, “lncRNA”, “Function” and “Reference”) and the body of the table (“Mammary tumor”, “Melanoma”, and so on…) to separate the fields.

Please write “Cancer types”, “lncRNA”, “Function” and “Reference” in bold.

Please replace “mammary tumor”, “melanoma”, and so on … to “Mammary tumor”, “Melanoma”, …

Response: Accepting your comments, we amend the table.

Reviewer 4 Report

lncRNA is a hot topic in cancer research. This article summarizes the function of lncRNAs as novel biomarkers in various types of canine tumors, which is meaningful for veterinary clinics. Some minor issues need to be addressed.

1. This paper is a little short, and it cannot meet the criteria of review, please rephrase the description of the abstract and summary.

2. From lines 67 to 70, these two statements are cited by the same reference, please add the citation number for both, and check the whole manuscript.

3. Lines 85 and 86 should cite a reference.

4. Lines 215 and 216, “Further investigations of these lncRNAs perhaps may amazing insights for veterinary clinics”. There are grammar errors, please correct them, and check the whole manuscript.

5. In part 3, there are only two references about lncRNAs in canine melanoma, it would be better to add more research.

6. In the title of Table 1, it would be better to change “canine cancer” to “canine tumor”, which will consist of the title of this manuscript.

Should be improved again.

Author Response

We deeply appreciate your critical comments, which have undoubtedly provided us with valuable opportunities to improve our work. We have revised the manuscript following the advice provided and addressed the issues raised by the reviewer. The changes were highlighted in yellow in the manuscript. The following are point-by-point responses to comments and suggestions from authors:

1. This paper is a little short, and it cannot meet the criteria of review, please rephrase the description of the abstract and summary.

Response: Accepting your comments, we rephrased the description of the abstract and summary

2. From lines 67 to 70, these two statements are cited by the same reference, please add the citation number for both, and check the whole manuscript.

Response: Accepting your comments, we have added the citation number, and checked our manuscript.

3. Lines 85 and 86 should cite a reference.

Response: Accepting your comments, we have added a citation.

4. Lines 215 and 216, “Further investigations of these lncRNAs perhaps may amazing insights for veterinary clinics”. There are grammar errors, please correct them, and check the whole manuscript.

Response: Accepting your comments, we rephrased this sentence.

5. In part 3, there are only two references about lncRNAs in canine melanoma, it would be better to add more research.

Response: The research on lncRNAs related to canine tumors is in an early stage, and there are a few studies we explore all the available resources. We are deeply sorry that we cannot add more references.

6. In the title of Table 1, it would be better to change “canine cancer” to “canine tumor”, which will consist of the title of this manuscript.

Response: Accepting your comments, we corrected it.

Round 2

Reviewer 2 Report

Dear Authors, the article is almost fine and will be ready for publication after very minimal refining for improving the communicability.

I detail the refinement as follow:

Line 176 Please, rephrase ... same mechanism in ...same mechanism in onco-pathology.

Line 215 Please, rephrase ... for veterinary medicine in ... for oncology or others pathological area in veterinary medicine.

Author Response

We deeply appreciate your comments, which undoubtedly provided valuable opportunities to improve our work. We have revised the manuscript following your advice. The changes were highlighted in yellow in the manuscript. The following are point-by-point responses to comments and suggestions from authors.

Reviewer 2

  1. Line 176 Please, rephrase ... same mechanism in ...same mechanism in onco-pathology.

Response: Accepting your comments, we replaced “... same mechanism in ...” with “same mechanism in onco-pathology”.

  1. Line 215 Please, rephrase ... for veterinary medicine in ... for oncology or others pathological area in veterinary medicine.

Response: Accepting your comments, we replaced “... for veterinary medicine in ...” with “for oncology or other pathological areas in veterinary medicine”